# One-Pot Synthesis and Evaluation of Antioxidative Stress and Anticancer Properties of an Active Chromone Derivative

**DOI:** 10.3390/molecules28073129

**Published:** 2023-03-31

**Authors:** Chirattikan Maicheen, Chokchaloemwat Churnthammakarn, Nichapat Pongsroypech, Thitiphong Khamkhenshorngphanuch, Jiraporn Ungwitayatorn, Kanin Rangsangthong, Rathapon Asasutjarit, Sewan Theeramunkong

**Affiliations:** 1Department of Pharmaceutical Chemistry, Faculty of Pharmaceutical Sciences, Huachiew Chalermprakiet University, Samut Prakan 10540, Thailand; 2Faculty of Pharmacy, Thammasat University, Pathumthani 12120, Thailand; 3Thammasat University Research Unit in Drug, Health Product Development and Application (DHP-DA), Department of Pharmaceutical Sciences, Faculty of Pharmacy, Thammasat University, Pathumthani 12120, Thailand; 4Department of Pharmaceutical Chemistry, Faculty of Pharmacy, Mahidol University, Bangkok 10400, Thailand

**Keywords:** chromone derivatives, anticancer, antioxidant, synthesis, quercetin

## Abstract

Chromones are the structural building blocks of several natural flavonoids. The synthesis of chromones, which contain a hydroxy group on the ring, presents some challenges. We used the one-pot method to synthesize ten chromone derivatives and two related compounds using modified Baker-Venkataraman reactions. The structures were confirmed using FT-IR, ^1^H NMR, ^13^C NMR, and HRMS. The in vitro antioxidant assay revealed that compounds **2e**, **2f**, **2j**, and **3i** had potent antioxidant activity and that all these synthesized compounds, except those containing nitro groups, were harmless to normal cells. In addition, compounds **2b**, **2d**, **2e**, **2f**, **2g**, **2i**, and **2j** had anticancer activity. Compounds **2f** and **2j** were used to investigate the mechanism of anticancer activity. Both **2f** and **2j** induced a slightly early apoptotic effect but significantly impacted the S phase in the cell cycle. The effect on cell invasion indicates that both compounds significantly inhibited the growth of cervical cancer cells. A chromone scaffold possesses effective chemoprotective and antioxidant properties, making it a promising candidate for antioxidant and future cancer treatments.

## 1. Introduction

Generally, human bodies produce damaging substances known as free radicals, which may also play a role in redox signaling and in regulating processes involving the maintenance of homeostasis when present in low-to-moderate concentrations [1]. When there is an excess of free radicals in the body, oxidative stress occurs and contributes to the development of various chronic and metabolic disorders or cancers [2]. If not neutralized, free radicals will initiate a chain reaction that can (a) cause structural modifications to cellular proteins, (b) stop the function of critical enzymes, cellular activities, and normal cell division, (c) degrade DNA, and (d) inhibit energy production [3,4,5,6,7]. In addition, the fact that free radicals are associated with aging implicates the gradual accumulation of oxidative cellular damage as a fundamental driver of cellular aging [8]. Furthermore, oxidative stress stimulates the immune response, causing allergic diseases and the expression of proinflammatory genes [9,10]. Several cancers are believed to result from interactions between free radicals and DNA, resulting in mutations that alter the cell cycle and cause neoplasia [11]. Healthy organisms resist the harmful effect of reactive oxygen species (ROS) due to the balance between oxidants and antioxidants [12,13]. Since tumor cells express fewer antioxidants than normal cells, their ROS levels are higher. Moreover, tumor cells with impaired mitochondrial oxidative metabolism exhibit elevated levels of ROS [14], making ROS induction a promising cancer strategy [15].

The antioxidant mechanism normally eliminates free radicals. Various endogenous and exogenous antioxidants protect against oxidative damage and chronic diseases [3]. Antioxidants can counteract free radicals by interfering with any of the three main stages of the oxidative process that are mediated by free radicals: initiation, propagation, and termination [16]. Flavonoids are an essential component in numerous nutraceutical, pharmaceutical, medicinal, and cosmetic applications. They have antioxidative, anti-inflammatory, antimutagenic, and anticarcinogenic properties as well as the ability to modulate key cellular enzyme functions [17,18], antifungal, antimicrobial [19], anti-parasitic properties [20]. Furthermore, flavonoids have potent inhibitory effects on several enzymes, including cyclo-oxygenase (COX), lipoxygenase, and phosphoinositide 3-kinase [17]. The flavone structure—also known as a phenyl-substituted chromone—has phenylbenzopyrone structures (C6-C3-C6) and is classified according to the saturation level and opening of the central pyran ring into flavones, flavanols, isoflavones, flavonols, flavanones and flavanonols (Figure 1) [21,22]. Notably, polyphenols have been reported to have anticarcinogenic properties against different cancers [23].

This study aimed to design novel compounds that enhance antioxidative activity and have the chemopreventive ability to prevent or reduce cervical cancer. The designed compounds—with chromones in the core structure (**2a**–**2j**, **3i**, and **4**)—were synthesized and tested for their antioxidative capacity compared to quercetin. Using modified Baker–Venkataraman reactions, a γ-pyrone structure was synthesized in a single pot. Acetophenones **1** was heated with a suitable acyl chloride in the presence of 1,8-diazabicyclo [5.4.0]undec-7-ene (DBU) in dry pyridine to produce the corresponding chromone derivative **2** and phenolic esters **3** in low yield, as shown in previous protocol (Figure 1) [24,25,26].

Our designed chromone compounds (**2a**–**2j**, **3i**, and **4**) were evaluated for their antioxidative capacity compared to the known antioxidative substance quercetin (Figure 1). The impacts of the hydroxyl group in relation to **2a**–**2f** were investigated, as well as variations in the hydroxyl group’s position in the A-ring and substituted withdrawing or donating groups that were visible in the C- and D-rings in **2g**–**2f**. In addition, the activity of the open B-ring was observed in relation to the significance of the chromone ring (**3**, **4**). At position 7, an ester substituent (**2i**) was also considered.

Three in vitro antioxidant assays (DPPH, FRAP, and ABTS) were utilized to evaluate the antioxidant property of the synthesized compounds. The DPPH (α, α-diphenyl-β-picrylhydrazyl) assay is the well-known approach to determine antioxidant activity using a stable free radical DPPH, measure the absorbance change, and calculate the absorbance change in the percentage of scavenging activity. For the FRAP (ferric reducing ability of plasma or ferric ion reducing antioxidant power) assay, this method does not involve radical generation or the scavenging of added radicals but uses a simple redox reaction. For example, the ferric-tripyridyltriazine (Fe^3+^-TPTZ, colorless) complex is reduced to ferrous tripyridyltriazine (Fe^2+^-TPTZ) at low pH and introduces a blue color with maximum absorption at 593 nm. The change in absorbance is related to antioxidant activity [27,28]. However, a significant limitation of these in vitro tests is the non-physiological measurement conditions (no enzymatic condition and no transition metal ion chelating activity); therefore, the results obtained may not reflect the antioxidant activity of the source material under the biological conditions that exist in humans and animals [29,30]. The ABTS assay involves a more drastic radical that is chemically produced. The use of ABTS for antioxidant activity determination of hydrophilic and lipophilic pure compounds has attracted interest due to its solubility in organic and aqueous media and stability in a wide pH range [31].

Furthermore, the cytotoxicity of the synthesized compounds was examined against both normal cells (HEK293 and HACAT cells) and cervical cancer cell lines (HeLa cell). Non-toxic and active compounds were chosen to further determine the protective activity against H_2_O_2_-induced cytotoxicity, then verify the outcome using a cellular ROS investigation. Cellular ROS in the cell was analyzed using probes that react with ROS to compete with antioxidants and produce a stable metabolite. Finally, anti-migration and apoptosis were also studied.

## 2. Results and Discussion

### 2.1. Synthesis

The desired acetophenone (1 eq) was reacted with acyl chloride (2.5 eq) in dry pyridine using DBU (3 eq) as the base. The chromone ring was produced at a high temperature (120–140 °C) for at least 24 h. The proposed mechanism of this reaction was illustrated in Figure 2. The reaction is firstly through hydroxy of phenyl reacting to acyl chloride and resulting non-chromone products occurred in some reactions. The crude product from synthesis was then purified by column chromatography. The product yield was obtained in the range of 5.2–66.2% (**2a**–**2j**), 24.9% (**3i**), and 17.5% (**4**). The lowest yield of the final product **2i** (5.2%) was due to the steric structure of the *nitro*- group, leading to the hindering of ring cyclization; additionally, the low yield of **2f** (7.3%) was possibly from the dihydroxy on the ring donate electron into the ring-affected C=O on the less reactive acetyl group and finally leading to low yield of the chromone ring. The synthesis protocol of **2j**, however, was the same as that of **2f,** except that 7,8-dihydroxy of the chromone ring can continue the reaction at 7-OH to form an ester group with a yield of 25.0%. During the reactions, the side products were compounds **3i** and **4**.

### 2.2. Antioxidant Activity

The in vitro antioxidant activity of the synthesized compounds was evaluated using the DPPH, FRAP, and ABTS assays. All techniques were modified and performed on 96-well plates. In addition, blank and quercetin were also performed, and the results are shown in Table 1.

**2f**, **2j,** and **3i** contained high in vitro antioxidant activity among these synthesized compounds. Without a chromone scaffold, compound **3i** still had high antioxidant activity. Noticeably, one part of the compound **3i** structure is similar to a simple structure of the natural protocatechuic acid (PCA), a well-known antioxidant.

### 2.3. Cytotoxicity Assay

Cell viability of cells treated with synthesized compounds was expressed as a percentage relative to untreated control cells (Table 2). Quercetin was used as a reference agent, which has been reported for its inhibitory activity against cervical cancer cells [32].

The results showed that most compounds were quite nontoxic in HEK293 cells, except compound **2i,** which contained *p*-NO_2_ and is likely to be more toxic after 24 and 48 h treatment. This is because the nitro group on the compound has a structural alert, which is extensively associated with toxicity. Furthermore, the synthesized compounds were generally more toxic in HACAT cells than in HEK293. The anticancer activity of these compounds on cervical cancer cells was also investigated for 24 h. Compounds **2b**, **2d**, **2e**, **2f**, **2g**, **2i**, and **2j** contained anticancer activity, and four compounds (**2i, 2b**, **2j**, **2e**) had the most potent anticancer activity with the IC_50_ values 34.9, 95.7, 101.0, and 107.6 μM, respectively (Table 3). Although the anticancer activity against HeLa cells of synthetic compounds seems to be weak compared to available anticancer agents, this scaffold has potential to be a preventive treatment for humans in the future. 

### 2.4. Antioxidant Activity against H_2_O_2_-Induced Cytotoxicity

The synthesized compounds were examined for the ability of the compounds to protect cells exposed to oxidative stress. Six active compounds from the antioxidant test, **2b**, **2d**, **2e**, **2f**, **2j**, and **3i**, were selected to study the ability of oxidative stress protection. The HEK293 cells were treated with 7.5 mM hydrogen peroxide for 30 min; samples and quercetin were added, and cell viability was investigated at 12 and 24 h treatment. The percentage of viable cells in each sample was calculated using the untreated cell as 100% survival (Figure 3).

Almost all compounds with 50 μM were able to significantly protect cells from oxidative stress at 24 h treatment (Figure 3). In particular, **2f** and **2j** at 100 μM were still able to protect cells from oxidative stress. Therefore, both compounds **2f** and **2j** have the potential for further investigation concerning in-cell ROS.

### 2.5. Cellular ROS Assay

ROS is the main contributor to cell death. These reactive molecules are formed by several mechanisms and can be detected by various techniques. In order to investigate the level of cellular ROS after treatment with compounds, the accumulation of intracellular ROS in cells was quantified with a cellular ROS assay kit (ab186029, Abcam, Boston, USA) fluorescence assay. After pretreatment with **2f**, **2j**, and **3i** at 50, 100, and 150 μM for 24 h, cell stress was induced with 5 mM hydrogen peroxide for 30 min. The cells in each well were collected and resuspended with a serum-free medium containing the ROS red probe. Subsequently, cells were incubated at 37 °C for 30 min in the dark, and fluorescence was read at Ex/Em = 650/675 nm. The fluorescence intensity of the cell after preincubation with **2f** and **2j** at 50, 100, and 150 μM was significantly reduced compared to the H_2_O_2_-treated cells (Figure 4). The fluorescence decreased significantly after pretreatment with **2f**, **2j,** and **3i,** and most of the fluorescence changes were among those with concentrations of 150 μM. Overall, these results suggested that pretreatment with **2f** and **2j** could reduce oxidative stress.

### 2.6. Apoptosis

Examining apoptosis induction by some samples was confirmed by the determination of annexin V (*p* ≤ 0.05) using the IC_50_ for **2b**, **2e**, **2f**, **2j**, and quercetin in the HeLa cell line (Figure 5a). In the apoptosis profile, the cells treated with **2f** were significantly changed (*p* < 0.05) compared to the control; −12.1% in the live phase, +6.3% in the early apoptosis phase, +1.2% in the late apoptosis phase, and +5.8% in the dead phase after 24 h treatment. However, other compounds had a lower effect on apoptosis than **2f**. The results of total apoptosis are presented in Figure 5b. The results reveal that compound **2f** induced apoptosis in tumor cells, and the mechanism that mediates the cytotoxicity of this compound was apoptosis induction.

### 2.7. Cell Cycle Analysis

In order to understand the influence of derivatives and quercetin on the cell cycle, HeLa cells were treated with **2f** and **2j** and quercetin at the IC_50_ for 24 h and analyzed by flow cytometry (Guava Muse Cell Analyzer, Luminex, USA) with Guava suite software 3.4. The results show S phase arrest in cell cycle after treatment with concentration at IC_50_ of quercetin and **2f**, **2j** for 24 h. The pattern of cell population in the G0/G1 and S phase of quercetin and **2f**, **2j** changed dramatically from the control (Figure 6). The paclitaxel (5 μM) was performed as the positive control. The pattern of the paclitaxel cell population was evident as an active inhibitor with potent phase arrest of G2/M. The percentage of cells in phase S was 13.79%, which clearly increased to 37.88%, 34.03%, and 31.61% in quercetin, **2f**, and **2j**, respectively (Table 4). This signifies that **2f** and **2j** arrested the cells in phase S to the same degree as quercetin.

### 2.8. Cell Migration Assay 

Cell migration in **2f** and **2j** compared to the untreated control were examined for 24 and 48 h. Quercetin was also investigated. In HEK293 cells, the migration of treated cells by **2f**, **2j**, and quercetin showed significant induction of cell migration compared to the untreated (Figure 7).

In addition, the effect of compounds on invasion inhibition in cancer cell lines was evaluated. The results indicated that **2f**, **2j**, and quercetin significantly inhibited the invasion of HeLa cells (Figure 8).

## 3. Materials and Methods

### 3.1. Apparatus, Chemical and Reagents

Commercially available reagents and solvents were used without further purification. Dry pyridine was drawn under a slightly positive atmosphere of dry nitrogen. The reactions were performed in flame- or oven-dried glassware under a positive dry nitrogen pressure when necessary. 

### 3.2. Synthesis

#### 3.2.1. General Procedure for the Synthesis 

In a stirred solution of acetophenone (0.5 g, 1 eq), dry pyridine (10 mL) was slowly added, followed by benzoyl chloride (2.5 eq) and DBU (3 eq). The reaction mixture was refluxed at 120 °C for 24 h, and the pyridine evaporated in vacuo. The mixture was extracted with ethyl acetate (2 × 50 mL). The combined organic layers were washed with water (2 × 50 mL), dried over anhydrous sodium sulfate, and filtered. After evaporation, the crude product was purified by column chromatography.

#### 3.2.2. Characterization of Synthesis Compounds

3-Benzoyl-5-hydroxy-2-phenyl-4H-chromen-4-one (**2a**): Orange solid; 0.746 g of (66.2%); melting point: 156.0–158.0 °C; Rf = 0.553 (hexane: ethyl acetate 7:3); IR 3728.40, 3339, 3063, 2930, 2857, 1678, 1649, 1472, 1229 cm^−1^; ^1^H-NMR (600 MHz, DMSO-d_6_, 25 °C) δ 12.08 (s, 1H), 8.01 (dd, *J* = 8.3, 1.2 Hz, 2H), 7.78 (t, *J* = 8.4 Hz, 1H), 7.67–7.61 (m, 3H), 7.53–7.44 (m, 5H), 7.25 (dd, *J* = 8.4, 0.7 Hz, 1H), 6.91 (dd, *J* = 8.3, 0.7 Hz, 1H). ^13^C-NMR (151 MHz, DMSO-d_6_, 25 °C) δ 192.69, 181.82, 163.86, 160.23, 156.60, 137.16, 134.79, 133.33, 132.38, 131.53, 129.80, 129.52, 129.38, 128.91, 121.26, 111.89, 110.21, 108.34; MS (ESI) *m*/*z* calculated: 342.0892 found: 343.0965 (M+H)^+^.

3-Benzoyl-6-hydroxy-2-phenyl-4H-chromen-4-one (**2b**): White solid; 0.210 g of (18.4%); melting point: 201.0–203.5 °C; Rf = 0.34 (hexane: ethyl acetate 7:3); IR 3331, 3063, 1900, 1676, 1468, 1359, 1127, 903 cm^−1^; ^1^H-NMR (600 MHz, DMSO-d_6_, 25 °C) δ 10.19 (br s, 1H), 7.97–7.85 (m, 2H), 7.70 (dd, *J* = 8.5, 0.9 Hz, 1H), 7.69–7.60 (m, 3H), 7.49–7.43 (m, 5H), 7.34 (m, 2H); ^13^C-NMR (151 MHz, DMSO-d_6_, 25 °C) δ 193.84, 175.94, 162.07, 157.39, 151.14, 137.07, 134.52, 132.05, 131.92, 129.65, 129.45, 129.29, 128.79, 124.57, 123.82, 121.83, 120.88, 107.89; MS (ESI) *m*/*z* calculated: 342.0892 found: 343.0964 (M+H)^+^.

3-Benzoyl-6-methoxy-2-phenyl-4H-chromen-4-one (**2c**): Yellow solid; 0.159 g of (13.5%); melting point: 194.0–196.0 °C; Rf = 0.683 (hexane: ethyl acetate 7:3); IR 3624, 3071, 2965, 2932, 2835, 1651, 1483, 1368, 1258 cm^−1^; ^1^H-NMR (600 MHz, DMSO-d_6_, 25 °C) δ 7.92 (d, *J* = 7.3 Hz, 2H), 7.79 (d, *J* = 9.2 Hz, 1H), 7.65–7.58 (m, 3H), 7.52 (dd, *J* = 9.2, 3.1 Hz, 1H), 7.48–7.43 (m, 6H), 3.88 (s, 3H). ^13^C-NMR (151 MHz, DMSO-d_6_, 25 °C) δ 193.84, 175.94, 162.07, 157.39, 151.14, 137.07, 134.52, 132.05, 131.92, 129.65, 129.45, 129.29, 128.79, 124.57, 123.82, 121.83, 120.88, 105.15, 56.32; MS (ESI) *m*/*z* calculated: 356.1049 found: 357.1119 (M+H)^+^.

3-Benzoyl-7-hydroxy-2-phenyl-4H-chromen-4-one (**2d**): White solid; 0.281 g of (24.8%); melting point: 270.0–271.0 °C; Rf = 0.53 (hexane: ethyl acetate 5:5); IR 3187, 3061, 1671, 1618, 1577, 1506, 1450, 1377, 1241 cm^−1^; ^1^H-NMR (600 MHz, DMSO-d_6_, 25 °C) δ 7.91 (dd, *J* = 8.4, 5.2 Hz, 3H), 7.63–7.58 (m, 3H), 7.48 (dd, *J* = 9.6, 5.9 Hz, 3H), 7.44 (d, *J* = 7.8 Hz, 2H), 7.02–6.98 (m, 2H); ^13^C-NMR (151 MHz, DMSO-d_6_, 25 °C) δ 193.95, 175.39, 163.90, 161.52, 158.17, 137.14, 134.45, 132.09, 131.78, 129.59, 129.43, 129.26, 128.71, 127.29, 122.17, 116.12, 115.74, 103.08; MS (ESI) *m*/*z* calculated: 342.0892 found: 343.0961 (M+H)^+^.

3-Benzoyl-5,7-dihydroxy-2-phenyl-4H-chromen-4-one (**2e**): Yellow solid; 0.090 g of (8.2%); melting point: 196.0–197.0 °C; Rf = 0.25 (hexane: ethyl acetate 7:3); IR 3204, 3011, 1628, 1588, 1459, 1369, 1280, 762 cm^−1^; ^1^H-NMR (600 MHz, DMSO-d_6_, 25 °C) δ 12.23 (s, 1H), 11.11 (s, 1H), 7.94 (d, *J* = 7.9 Hz, 2H), 7.65–7.55 (m, 3H), 7.47–7.41 (m, 5H), 6.53 (s, 1H), 6.28 (s, 1H). ^13^C-NMR (151 MHz, DMSO-d_6_, 25 °C) δ 192.90, 180.30, 165.51, 162.76, 161.91, 158.07, 137.01, 134.69, 132.15, 131.60, 129.71, 129.50, 129.32, 128.78, 120.68, 103.84, 99.89, 94.87; MS (ESI) *m*/*z* calculated: 358.0841 found: 359.0913 (M+H)^+^.

3-Benzoyl-7,8-dihydroxy-2-phenyl-4H-chromen-4-one (**2f**): Yellow solid; 0.091 g of (7.3%); melting point: 221.5–223.0 °C; Rf = 0.27 (hexane: ethyl acetate 4:6); IR 3112, 3009, 2956, 1652, 1557, 1448, 1368, 1268, cm^−1^; ^1^H-NMR (600 MHz, DMSO-d_6_, 25 °C) δ 10.54 (s, 1H), 9.57 (s, 1H), 7.89 (dd, *J* = 14.0, 1.2 Hz, 2H), 7.69–7.57 (m, 3H), 7.55–7.38 (m, 6H), 7.02 (d, *J* = 8.7 Hz, 1H). ^13^C-NMR (151 MHz, DMSO-d_6_, 25 °C) δ 194.19, 175.92, 161.34, 151.58, 147.27, 137.21, 134.40, 133.69, 132.19, 131.75, 129.57, 129.43, 129.19, 128.88, 121.59, 116.51, 115.81, 114.99; MS (ESI) *m*/*z* calculated: 358.0841 found: 359.0910 (M+H)^+^.

7-Hydroxy-3-(3-methoxybenzoyl)-2-(3-methoxyphenyl)-4H-chromen-4-one (**2g**): Yellow solid; yield 38.6%; melting point: 253.0–254.0 °C; Rf = 0.48 (hexane: ethyl acetate 5:5); IR 3046, 2887, 1680, 1627, 1609, 1583, 1459, 1233, cm^−1^; ^1^H-NMR (600 MHz, DMSO-d_6_, 25 °C) δ 10.97 (s, 1H), 7.96–7.72 (m, 3H), 7.63–7.51 (m, 2H), 7.06–6.90 (m, 5H), 3.82 (s, 3H), 3.76 (s, 3H); ^13^C-NMR (151 MHz, DMSO-d_6_, 25 °C) δ 193.65, 175.39, 163.84, 161.14, 160.02, 159.50, 158.12, 138.56, 133.32, 130.70, 130.58, 127.26, 122.60, 122.24, 120.94, 120.52, 117.35, 116.09, 115.78, 114.23, 113.43, 103.14, 55.86, 55.61; MS (ESI) *m*/*z* calculated: 402.1103 found: 403.1165 (M+H)^+^.

7-hydroxy-3-(4-methoxybenzoyl)-2-(4-methoxyphenyl)-4H-chromen-4-one (**2h**): Yellow solid; yield 28.0%; melting point: 300.0–301.0 °C; Rf = 0.63 (hexane: ethyl acetate 5:5); IR 3108, 3075, 2957, 1662, 1606, 1506, 1456, 1262, 1029, cm^−1^; ^1^H-NMR (600 MHz, DMSO-d_6_, 25 °C) δ 11.01 (s, 1H), 7.91 (d, *J* = 8.6 Hz, 1H), 7.52–7.48 (m, 1H), 7.42–7.33 (m, 3H), 7.23–7.19 (m, 1H), 7.19–7.15 (m, 1H), 7.15–7.13 (m, 1H), 7.08–7.04 (m, 1H), 7.03–6.98 (m, 2H), 3.78 (s, 3H), 3.67 (s, 3H); ^13^C-NMR (151 MHz, DMSO-d_6_, 25 °C) δ 192.48, 175.40, 164.19, 163.66, 161.93, 160.74, 158.01, 132.06, 130.39, 130.35, 127.22, 124.24, 121.28, 115.86, 115.71, 114.74, 114.68, 103.00, 56.06, 55.87; MS (ESI) *m*/*z* calculated: 402.1103 found: 403.1166 (M+H)^+^.

7-hydroxy-3-(4-nitrobenzoyl)-2-(4-nitrophenyl)-4H-chromen-4-one (**2i**): Yellow solid; yield 5.2%; melting point: 290.0–291.0 °C; Rf = 0.54 (hexane: ethyl acetate 5:5); IR 3225, 3108, 1680, 1621, 1524, 1453, 1347, 1238, 1109 cm^−1^; ^1^H-NMR (600 MHz, DMSO-d_6_, 25 °C) δ 11.26 (s, 1H), 8.31–8.26 (m, 4H), 8.24–8.20 (m, 2H), 7.91 (d, *J* = 8.7 Hz, 1H), 7.88–7.84 (m, 2H), 7.06 (dd, *J* = 15.5, 5.0 Hz, 2H); ^13^C-NMR (151 MHz, DMSO-d_6_, 25 °C) δ 192.48, 175.40, 164.19, 163.66, 161.93, 160.74, 158.01, 132.06, 130.39, 130.35, 127.22, 124.24, 121.28, 115.86, 115.71, 114.74, 114.68, 103.00; MS (ESI) *m*/*z* calculated: 432.0594 found: 433.0651 (M+H)^+^.

3-Benzoyl-8-hydroxy-4-oxo-2-phenyl-4H-chromen-7-yl benzoate (**2j**): Orange solid; 0.2686 g of (25.0%); melting point: 205.0–206.0 °C; Rf = 0.125 (hexane: ethyl acetate 7:3); IR 3163.26, 1741.72, 1450.47, 1386.84, 1328.95cm^−1^; ^1^H-NMR (600 MHz, DMSO-d_6_, 25 °C) δ 11.43 (s, 1H), 8.22 (t, *J* = 7.0 Hz, 2H), 7.92 (d, *J* = 8.1 Hz, 2H), 7.87 (d, *J* = 8.9 Hz, 1H), 7.81–7.76 (m, 1H), 7.68–7.60 (m, 3H), 7.51–7.46 (m, 2H), 7.41 (d, *J* = 8.0 Hz, 2H), 7.39–7.34 (m, 1H), 7.27 (t, *J* = 7.8 Hz, 2H), 7.22 (d, *J* = 8.9 Hz, 1H). ^13^C-NMR (151 MHz, DMSO-d_6_, 25 °C) δ 193.65, 175.23, 164.31, 160.91, 155.74, 150.14, 137.01, 134.93, 134.62, 132.04, 131.64, 130.49, 129.62, 129.51, 129.44, 129.28, 128.55, 128.29, 126.24, 123.67, 122.25, 116.22, 116.09; MS (ESI) *m*/*z* calculated: 462.1103 found: 463.1175 (M+H)^+^.

6-Acetyl-2,3-dihydroxyphenyl benzoate (**3i**): White solid; 0.2251 g of (24.9%); melting point: 166.5–168.0 °C; Rf = 0.52 (hexane: ethyl acetate 5:5); IR 3096, 2930, 1742, 1283, 1260, 1067, 700 cm^−1^; ^1^H-NMR (500 MHz, DMSO-d_6_, 25 °C) δ 12.81 (s, 1H), 11.04 (s, 1H), 8.12 (d, *J* = 7.2 Hz, 2H), 7.74 (t, *J* = 7.4 Hz, 2H), 7.60 (t, *J* = 7.7 Hz, 2H), 6.58 (d, *J* = 8.9 Hz, 1H), 2.57 (s, 3H). ^13^C-NMR (151 MHz, DMSO-d_6_, 25 °C) δ 204.06, 164.06, 157.06, 156.43, 134.47, 130.57, 130.35, 129.42, 129.11, 126.11, 113.69, 108.37, 26.85; MS (ESI) *m*/*z* calculated: 254.0685 found: 273.0757 (M+H)^+^.

2-Acetyl-5-hydroxyphenyl furan-2-carboxylate (**4**): White solid; 0.1969 g of (17.5%); melting point: 122.0–124.0 °C; Rf = 0.80 (hexane: ethyl acetate 5:5); IR 3102, 3066, 2990, 1755, 1263, 1045, 704; ^1^H-NMR (600 MHz, DMSO-d_6_, 25 °C) δ 12.20 (s, 1H), 8.13 (dd, *J* = 1.6, 0.7 Hz, 1H), 8.00 (d, *J* = 8.6 Hz, 1H), 7.61 (dd, *J* = 3.6, 0.7 Hz, 1H), 6.97–6.87 (m, 2H), 6.82 (dd, *J* = 3.6, 1.7 Hz, 1H), 2.66 (s, 3H). ^13^C-NMR (151 MHz, DMSO-d_6_, 25 °C) δ 204.04, 162.48, 156.00, 155.81, 149.43, 143.08, 133.44, 121.22, 119.22, 113.66, 113.36, 111.08, 28.27; MS (ESI) *m*/*z* calculated: 246.0528 found: 247.0601 (M+H)^+^. (Appendix A: ^1^H-NMR spectrum of compound **2a**–**3i**; Appendix A: ^1^H-NMR spectrum of compound **4**; Appendix A: ^13^C-NMR spectrum of compound **2a**–**3i**; Appendix A: ^13^C-NMR spectrum of compound **4**; Appendix A: Mass spectrum (MS-ESI) of com-pound **2a**–**3i**; Appendix A: Mass spectrum (MS-ESI) of compound **4**).

### 3.3. Antioxidant Activity 

#### 3.3.1. DPPH Assay 

The DPPH method was followed and modified from a previous study [33]. Trolox was used as a reference. The DPPH stock solution at a concentration of 0.625 mM was prepared in 50% ethanol and kept at 4 °C in the dark. On the day of the experiment, 0.208 mM fresh DPPH working solution was freshly prepared by further diluting with 50% ethanol. Trolox and sample stock solutions were prepared in a serial dilution with 50% ethanol at concentrations ranging from 12.5–750 μM and stored at 4 °C. Each reaction was carried out in a 96-well plate, and 100 μL of test preparation and DPPH solution were added. The 96-well plate was then shaken well for 10 min and kept in the dark at room temperature for 20 min, and the absorbance was recorded at 515–517 nm. According to the method described, delocalization also results in a deep violet color and is quantified as a percentage of scavenging activity.
% scavenging activity (SA) = 100 × (A_sample_ − A_blank_)/(A_control_ − A_blank_)(1)
where A_control_ and A_sample_ are the absorbances of the test DPPH without and with a sample, respectively. A_blank_ is the absorbance of the solvent used. The EC_50_ values were calculated from the relationship curve of %SA versus sample concentration. The experiment was carried out at least in triplicate. 

#### 3.3.2. FRAP Assay

The FRAP assay is a method for assessing antioxidant power. Briefly, the FRAP reagent was prepared from acetate buffer pH 3.6, 10 mM TPTZ solution in 40 mM HCl and 20 mM FeCl_3_.6 H_2_O in relative proportions of 10:1:1, *v*/*v*, respectively. The FRAP reagent was freshly prepared daily. A solution of 5 μL of the sample (3–300 μM, *approx.*) was mixed with 180 μL of the FRAP reagent in a 96-well plate. The mixture was shaken in the dark for 10 min and incubated at 37 °C for 20 min, and the absorbance was determined at 593 nm. The solution without the sample was used as a blank. All determinations were made in triplicate. Trolox was used as a standard antioxidant and plot the calibration curve between absorbance and concentration. Quercetin was applied as the reference compound. The FRAP values were calculated by comparing the absorbance change in the test sample and represented in μM of Trolox equivalent per μM of sample.

#### 3.3.3. ABTS Assay

The ABTS assay was prepared by reaction of 5 mL of a 7 mM aqueous ABTS (2,2′-azino-bis (3-ethylbenzthiazoline-6-sulphonic acid) solution and 2.45 mM potassium persulfate solution (K_2_S_2_O_8_) [34]. After preparation and storage in the dark for 16 h, the radical cation solution was further diluted in ethanol to absorb about 0.7 at 734 nm. Trolox and test sample solutions were prepared in stock solution with DMSO and diluted with ethanol. A solution of 100 μL of the sample was mixed with 100 μL of the ABTS reagent in a 96-well plate. The mixture was shaken in a dark place for 10 min and incubated at 37 °C for 20 min, and the absorbance was determined at 734 nm. Absorbance determinations were recorded immediately and not more than 1 h after adding the reagent. The solution without the sample was used as a blank. All the determinations were made in triplicate. The sample’s scavenging percentage was calculated from this equation: scavenging percentage = (Absorbance _control_ − Absorbance _sample_) × 100/Absorbance _control_. EC_50_ values were calculated from the relationship curve of %scavenging versus sample concentration.

### 3.4. Cell Culture

HEK293, HACAT, and HeLa cells were grown in modified Eagles’ minimum essential medium (EMEM, Gibco) with 10% fetal bovine serum (FBS, Gibco), 1% of 100 U/mL streptomycin and 100 U/mL penicillin (Gibco). Cells were seeded in culture flasks and maintained in an incubator at 37 °C in a humidified atmosphere with 5% CO_2_. After reaching 80–90% of confluence, cells were detached by incubation with 0.25% trypsin-0.2% EDTA (Gibco) for 5 min and plated for experiments. 

### 3.5. Cytotoxicity Assay

This study examined cell cultures of HEK293, HACAT, and HeLa cells. The cytotoxicity of the test samples was carried out by the MTT (3-(4,5-dimethylthiazol-2-yl)-2,5-diphenyltetrazolium bromide) assay. The cells were seeded in 96-well microliter plates (2.0 × 10^4^ cells/well). Stock solutions of the test compounds were prepared in DMSO. After 24 h, cells were treated with different concentrations of samples and diluted with media to obtain the final desired concentration. The treated cells were then incubated at 37 °C and 5% CO_2_ for 24 and 48 h. After incubation, the medium was replaced with 0.05% *w*/*v* of MTT solution in PBS and further incubated for 3 h. The solution was then discarded and DMSO was added to dissolve the formed formazan crystals. Each well’s absorbance (A) was measured at 570 nm using a microplate reader (FLUOstar Omega^TM^, BMG Labtech, Ortenberg, Germany). All experiments were carried out in triplicate and repeated for at least three experiments.

### 3.6. Cellular ROS Assay

In this study, Hela cells were seeded overnight at 20,000 cells/90 µL/well in a black wall/clear bottom 96-well plate. Cells included those untreated (control) and those pre-treated with compounds at various concentrations (50, 100, 150 μM) for 24 h. The ROS deep red assay solution (100 μL/well) was added and incubated at 5% CO_2_, 37 °C incubator for 30 min. The fluorescence signal was monitored at Ex/Em = 650/675 nm (cut off = 665 nm) with bottom read mode within 30 min (cellular ROS assay kit, ab186029, abcam).

### 3.7. Apoptosis

HeLa cells were seeded overnight at 1 × 10^5^ cells/1.5 mL/well in a 6-well plate. The cells were treated with a sample with IC_50_ concentrations and incubated at 37 °C and 5% CO_2_ for 24 h. The cells were lyzed and washed with PBS. The collected cells were diluted with serum-free medium to a suitable volume. The 100 μL of cell suspension was added to 100 μL of Muse^TM^ Annexin V & Dead Cell reagent and incubated for 20 min at room temperature. Cells were analyzed by flow cytometry (Guava Muse Cell Analyzer, Luminex, Austin, TX USA) with Guava suite software 3.4.

### 3.8. Cell Cycle Analysis

The treatment of HeLa cells and compounds was the same as in the apoptosis protocol. The cells were harvested for 200 μL and centrifuged at 600× *g* for 5 min and washed with PBS. The mixing cells were slowly added 200 μL of ice-cold 70% ethanol and incubated for at least 3 h at −20 °C. Cells were centrifuged and washed with PBS. Finally, cells were stained with 200 μL of Muse^®^ cell cycle reagent and incubated at room temperature for 30 min in the dark. The cell was analyzed by flow cytometry (Guava Muse Cell Analyzer, Luminex, Austin, TX, USA).

### 3.9. Cell Migration Assay

Cell migration in the HEK293 and the HeLa cell lines was investigated. The assay was conducted to investigate cell maintenance after being damaged. The active samples from in vitro antioxidant activity were examined. Cells were grown in a 24-well plate until 80% confluence and treated with **2f** and **2j** for 24 or 48 h compared to the untreated control. The cells in each well were then scratched to create a wound area across the center of the plates using a plastic 20 μL pipette tip. Cell images were taken immediately, and the scratch area was measured at 0, 24, and 48 h. During the experiment, the percentage of wound area was tracked as an indirect assessment. Compared to the control, the percentage of empty area change at each time point was quantified with a Nikon eclipse Ts2R inverted microscope.

## 4. Conclusions

In summary, all compounds were synthesized and screened for antioxidant activity. Four compounds—**2e**, **2f**, **2j,** and **3i**—displayed potent antioxidant activity and were safe for normal cells. In addition, all synthesis compounds were investigated for their anticancer activity in cervical cancer cells. Compounds **2b**, **2d**, **2e**, **2f**, **2g**, **2i**, and **2j** also showed activity against cancer, but **2i** exhibited toxicity in normal cells. The compounds **2f** and **2j** at concentration of 50 and 100 μM are observed to protect cells after treatment with hydrogen peroxide. These two compounds were likely to be lead compounds. From the cytotoxicity test and anti-migration in HeLa, compounds **2f** and **2j** exhibited potential chemopreventive properties. Furthermore, **2f** and **2j** can induce cell cycle arrest in the S phase and induce cell apoptosis in HeLa cells. These results encourage the use of these chromone scaffolds as a promising antioxidant and chemopreventive agent. All authors have read and agreed to the published version of the manuscript.

## Data Availability

Not applicable.

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
