# Peer review of "One-Pot Synthesis and Evaluation of Antioxidative Stress and Anticancer Properties of an Active Chromone Derivative"

_molecules, 2023, doi:10.3390/molecules28073129_

Round 1
Reviewer 1 Report
The manuscript "One-Pot Synthesis and Evaluation of Antioxidative Stress and Anticancer Properties of an Active Chromone Derivative" by Theeramunkong et al. describes the synthesis and biological evaluation of a small library of chromones obtained through a Baker-Venkataraman approach, resulting in different functionalizations of the chromone ring and phenyl substituent. Antioxidant and anticancer properties have been evaluated for these products.
Using various acetophenones and benzoyl chlorides, the authors derived chromone derivatives in moderate/low yields using a simple and rapid approach; although the synthesis does not appear to be very novel and the yields are not astounding, the idea sounds intriguing and good; the overall synthesis is streamlined and straightforward, and the characterization of the compounds is thorough and comprehensive.
In order to understand the biological properties of the compounds synthesized, a deep biological evaluation was performed, resulting in the identification of four different compounds as possible safe antioxidants, and two of them as potential candidates for further optimization in the development of novel anticancer and chemopreventive agents.
I found it to be a well-structured manuscript, with solid and coherent data supporting all of the conclusions reached. It is a well-written document that is easy to understand.
Therefore, I recommend publication in Molecules after the following minor corrections:
- Page 3: The last sentence before scheme 1 . "In particular, acetophenones without an OH group at the orthoposition did not produce the desired chromone." should be deleted. O-hydroxy groups are essential for the reaction, so this sentence is unnecessary.
- Page 4, chapter 2.1: Please change "The lowest yield of the final product (2f, 7.3%)" to "The lowest yield in the synthesis of product 2f (7.3%)". It would be helpful if you could provide a rationale for why the dihydroxy moiety results in a lower yield.
- Compound 4 shows an interesting furan ring. Can the authors provide any insight into how this product is formed?
- Chapter 3.2.2: Correct with Capital letter
- Chapter 3.2.2: Compounds 2d and 2g do not have characterizations. Even if the compounds have already been published, please provide characterization and yield information or a reference for comparison.
- Carefully check the font dimension in all the section of the manuscript
Author Response
- Page 3: The last sentence before scheme 1. "In particular, acetophenones without an OH group at the ortho-position did not produce the desired chromone." should be deleted. O-hydroxy groups are essential for the reaction, so this sentence is unnecessary.
Reply: We already remove the sentence.
- Page 4, chapter 2.1: Please change "The lowest yield of the final product (2f, 7.3%)" to "The lowest yield in the synthesis of product 2f (7.3%)". It would be helpful if you could provide a rationale for why the dihydroxy moiety results in a lower yield.
Reply: We amended the manuscript
- Compound 4 shows an interesting furan ring. Can the authors provide any insight into how this product is formed?2
Reply: We are sorry for my too shortcut of figure. We redraw new separated reaction to clarify the furan ring product.
- Chapter 3.2.2: Correct with Capital letter
Reply: We amended the manuscript
- Chapter 3.2.2: Compounds 2d and 2g do not have characterizations. Even if the compounds have already been published, please provide characterization and yield information or a reference for comparison.
Reply: We already add characterization of all compounds.
- Carefully check the font dimension in all the section of the manuscript
Reply: We amended the manuscript
Reviewer 2 Report
Before their work is accepted for publication in Molecules, I suggest the authors to address the following issues.
In scheme 1, the synthesis of compound 4 is not clear. Authors could draw a separate reaction for its synthesis.
In scheme 1, the reagent and conditions have not been completely presented. Please add the conditions above the reactions arrow or in the scheme caption.
In “Results and Discussion” section for Chemistry part, a lot of experimental details were mentioned. Such details should be moved to the Experimental section.
In Section “2.3. Cytotoxicity Assay”: Authors should mention that the obtained anticancer activity against HeLa cells is weak activity.
Authors stated “three compounds (2b, 2f, 2j) had the most potent anticancer activity with the IC50 values 95.7, 101.0, and 107.6 M, respectively”, whereas, compound 2i was found to be the most active counterpart in this assay. Please correct.
In the Experimental section, the physical and spectral data for many compounds are missing.
The provided ranges for the melting points are very broad and are unacceptable. Please revise and re-measure the melting point for all compounds.
The quality of the graphical abstract should be enhanced.
Please revise the NMR characterization.
There are some typos and language mistakes, please revise thoroughly.
Author Response
- In scheme 1, the synthesis of compound 4 is not clear. Authors could draw a separate reaction for its synthesis.
Reply: We amended the manuscript
- In scheme 1, the reagent and conditions have not been completely presented. Please add the conditions above the reactions arrow or in the scheme caption.
Reply: We amended the manuscript
- In “Results and Discussion” section for Chemistry part, a lot of experimental details were mentioned. Such details should be moved to the Experimental section.
Reply: We let’s try to move but we discuss that the manuscript seem to be incomplete for some part and non-consistency, So, we design to remain core part as previous.
- In Section “2.3. Cytotoxicity Assay”: Authors should mention that the obtained anticancer activity against HeLa cells is weak activity.
Reply: We amended the manuscript
- Authors stated “three compounds (2b, 2f, 2j) had the most potent anticancer activity with the IC50 values 95.7, 101.0, and 107.6 mM, respectively”, whereas, compound 2i was found to be the most active counterpart in this assay. Please correct.
Reply: We amended the manuscript
- In the Experimental section, the physical and spectral data for many compounds are missing.
Reply: We already add the manuscript
- The provided ranges for the melting points are very broad and are unacceptable. Please revise and re-measure the melting point for all compounds.
Reply: We are apologized for these things. We reexamined and amended the manuscript
- The quality of the graphical abstract should be enhanced.
Reply: We adjust the resolution of graphical abstract
- Please revise the NMR characterization.
Reply: We amended and fill up the manuscript
- There are some types and language mistakes, please revise thoroughly.
Reply: We amended the manuscript
Reviewer 3 Report
The synthesis of chromone derivatives has been described. The authors have used one-pot synthesis with moderate yields. In addition, some derivatives have been shown to have antioxidant activity and activity against cancer.
This work should be improved according to the following comments:
1) Check the codes of compounds, they should be in bold. Also, check the style (text formatting, font size, etc.).
3) Please, add the yields for the compounds in Scheme 1. Correct “5,7-di(OH)2” it should be “5,7-di(OH)”, same for “7,8-di(OH)2”.
4) Specify why only 2f was converted to 2j.
5) The possible mechanism of the reaction may be useful to readers.
6) In 3.2.2. characterization of synthesis compounds: correct the yields for all compounds “0.090 g of (8.2%)”.
7) It is necessary to correct typos that occur in the text.
8) Clarify, how did you prove the structure of ortho-substituted isomer 3i.
Author Response
- Check the codes of compounds, they should be in bold. Also, check the style (text formatting, font size, etc.).
Reply: We amended the manuscript
- Please, add the yields for the compounds in Scheme 1. Correct “5,7-di(OH)2” it should be “5,7-di(OH)”, same for “7,8-di(OH)2”.
Reply: We amended the manuscript
- Specify why only 2f was converted to 2j.
Reply: In fact, not only 2f converted to 2j, the previous work we met compound 2d, 2g, 2h and 2i (containing OH at position 7) can convert R-OH to R-COOR¢ (ester) and we found that all of them were inactive in both in vitro antioxidant assay and in cell assay, that is why we did not mention here. Therefore, our group further attempted to find the best compounds that should be potential products and found this compound 2f and 2j were quite active.
4) The possible mechanism of the reaction may be useful to readers.
Reply: We amended the manuscript
5) In 3.2.2. characterization of synthesis compounds: correct the yields for all compounds “0.090 g of (8.2%)”.
Reply: We amended the manuscript
6) It is necessary to correct types that occur in the text.
Reply: We amended the manuscript
7) Clarify, how did you prove the structure of ortho-substituted isomer 3i.
Reply: The reasons are
- In previous study, Dr. Maicheen and Dr. Ungwitayatorn already synthesized lead compound 2d and they confirmed the structure with COSY (2D-NMR) technique and see the proton should be occur at OH in o-position. (see figure as attached file)
- We found the several studies displayed the confirmation that cyclization reaction was happen at ortho position. [Ungwitayatorn, J.;2011, Chee, C.F 2011, Riva, C.1997]
- Furthermore, our synthesized compounds were also investigate with other characterization techniques and found they are corresponding to their structures, so, our group quite confirmed that our compounds were displayed as we purposed.
Reference
- Ungwitayatorn, J.; Wiwat, C.; Samee, W.; Nunthanavanit, P.; Phosrithong, N. Synthesis, in vitro evaluation, and docking studies of novel chromone derivatives as HIV-1 protease inhibitor. Mol. Struct. 2011, 1001(1-3), 152-161.
- Chee, C.F.; Buckle M.J.C.; Rahman, N.A. An efficient one-pot synthesis of flavones. Tetrahedron Lett. 2011, 52, 3120-3123.
- Riva, C. De.; Toma, C,; Donadd, L.;, Boi, 0.;C, Pennini, R.; Motta, G.; Leonardi, A, New DBU (1,8-diazabicyclo[5.4.0]undec-7-ene) assisted one-pot synthesis of 2,8-disubstituted 4H-1-benzopyran-4-ones. Synth. 1997, 195–201.

Round 2
Reviewer 2 Report
The manuscript has been improved, and should be accepted now.